# Anticandidal and Antibiofilm Effect of Synbiotics including Probiotics and Inulin-Type Fructans

**DOI:** 10.3390/antibiotics11081135

**Published:** 2022-08-21

**Authors:** Ricardo García-Gamboa, Miguel Domínguez-Simi, Misael S. Gradilla-Hernández, Jorge Bravo, Andrés Moya, Blanca Ruiz-Álvarez, Marisela González-Avila

**Affiliations:** 1Centre for Research and Assistance in Technology and Design of the State of Jalisco A.C., Normalistas 800, Guadalajara 44270, Mexico; 2Tecnológico de Monterrey, Escuela de Ingeniería y Ciencias, Av. General Ramón Corona 2514, Nuevo Mexico, Zapopan 45138, Mexico; 3Foundation for the Promotion of Health and Biomedical Research in the Valencian Community Public Health (FISABIO), 46020 Valencia, Spain

**Keywords:** biofilms, *Candida*, probiotics, inulin-type fructans, antimicrobial activity, antibiofilm effect, prebiotics

## Abstract

Background: There is great interest in the search for new alternatives to antimicrobial drugs, and the use of synbiotics is a promising approach to this problem. This study evaluated the growth inhibition and antibiofilm activity of the short-chain fatty acids produced by *Lacticaseibacillus rhamnosus* and *Pediococcus acidilactici* in combination with inulin-type fructans against *Candida albicans*. Methods: The growth inhibition of *Candida* was evaluated using microdilution analysis in 96-well microtiter plates; different concentrations of cell-free supernatants of *Lacticaseibacillus rhamnosus* and *Pediococcus acidilactici* were exposed to *Candida albicans*. The antibiofilm assessment was carried out using the crystal violet staining assay. The short-chain fatty acids were analyzed by gas chromatography. Results: The clinically isolated *Candida albicans* interacted with supernatants from *Lacticaseibacillus rhamnosus* and *Pediococcus acidilactici* and showed significant growth inhibition and antibiofilm formation versus the controls. Lactate and acetic acid were elevated in the supernatants. The results suggest that the supernatants obtained from the synbiotic combinations of *Lacticaseibacillus rhamnosus* and *Pediococcus acidilactici* with inulin-type fructans can inhibit the growth and biofilm formation against a clinically isolated *Candida albicans* strain. Conclusions: These results suggest that synbiotic formulations could be a promising alternative to antifungal drugs in candidiasis therapy.

## 1. Introduction

The World Health Organization (WHO) defines probiotics as live microorganisms that confer benefits on the health host [1]. Probiotics can prevent and treat infectious diseases and inflammation and can reduce the development of harmful pathogens [2,3,4]. Prebiotics are used to enhance probiotic properties [5,6]. *Lactobacillus* and *Bifidobacterium* are the most commonly used genera as probiotics [7,8]. *Lacticaseibacillus rhamnosus* [9] and *Pediococcus acidilactici* [10] have shown antimicrobial properties.

*Candida* spp. are opportunistic fungal pathogens that can cause mycosis in humans [11]. The clinical manifestations of *Candida* spp. range from localized to invasive and systemic disease. The disease varies depending on the patient’s immune status [12]. *Candida albicans* is the species most commonly associated with human fungal infections with a high mortality rate [13]. *Candida albicans* is part of the normal human microbiota on the mucous surfaces of the oral cavity, the gastrointestinal tract [14], and the vagina [15]. Several factors intervene in the transition of *Candida albicans* from a commensal to a pathogenic fungus: the expression of several virulence factors of *Candida albicans* leading to filament and biofilm formation as well as the synthesis of adhesin factors and related enzymes [16].

Furthermore, one of the great challenges to the use of conventional antimicrobials is the multifactorial nature of biofilm development, as it indicates the need for multi-targeted or combinatorial treatment. While azoles, polyenes, and echinocandins are common antifungal drugs, they have some adverse effects, such as mutation and drug resistance with *Candida albicans* [17]. Several studies have focused on the development of antimicrobials against bacteria [18] and have shown better results when the antimicrobial compounds have been functionalized into nanoparticles that mimic extracellular structures as release vectors for antibacterial agents [19]. Moreover, Hong et al., 2019 [20] reported that pullulan nanoparticles enhanced the antibacterial properties of *Lactobacillus plantarum* because this probiotic produced more plantaricin, a natural antibacterial peptide. In the literature, substances with growth inhibition activity have also been reported for fungal microorganisms. A previous study suggested that *Lacticaseibacillus rhamnosus* can alter the susceptibility of *Candida albicans* to antifungal drugs [12]. Another study demonstrated that cell-free supernatants (CFS) obtained from *Pediococcus acidilactici* inhibited *Candida albicans* [21]. Moreover, Fatty acids exerted through various physical, chemical, or biological mechanisms possess anti-infective activities; recently, several researchers have directed their interests towards the study of antimicrobial lipids and have demonstrated that fatty acids inhibit or kill a wide spectrum of pathogens, suppress the expression of Quorum Sensing-regulated genes, reduce swarming motility, adhesion, and virulence, and directly induce biofilm dispersion [22].

Inulin-type fructans are a prebiotic that can enhance the properties of beneficial intestinal bacteria [23,24]. However, there is a lack of studies focusing on the enhancement of fructans on the antifungal effect against *Candida albicans*. This study assessed the growth inhibition and antibiofilm effect of cell-free supernatants obtained from a synbiotic combination of *Lacticaseibacillus rhamnosus* and *Pediococcus acidilactici* supplemented with inulin-type fructans.

## 2. Results

Table 1 shows all of the CFS obtained from the probiotics. CFS from *Lacticaseibacillus rhamnosus* and *Pediococcus acidilactici* were exposed to *Candida* spp. to assess their growth inhibitory effect. Figure 1a,b shows the kinetic growth of *Candida* spp. in the presence of different CFS. The CFS-50%-W-D-*rhamnosus* (O.D. 0.71 ± 0.007) inhibited the growth of *Candida albicans* ATCC 10231 versus the control (O.D. 1.217 ± 0.06) (*p <* 0.005). The presence of CFS-25%-W-D-*rhamnosus*, CFS-50%-W-D-*rhamnosus*, and CFS-50%-W-I-*rhamnosus* inhibited growth against clinical *C. albicans* (O.D. 0.081 ± 0.003, 0.804 ± 0.03 and 0.870 ± 0.016, respectively) (*p <* 0.05). This inhibitory effect diminished when these supernatants were neutralized, as shown in Figure 1c,d. The inhibition effect persisted when the *L. rhamnosus* supernatants were submitted to thermal treatment. The 12.5, 25, and 50% concentrations of dextrose and inulin supernatants inhibited the growth of clinical *C. albicans* (Figure 1f), suggesting that minimal amounts of supernatants showed inhibition, thus showing a statistical difference (*p <* 0.05) versus controls.

Figure 2 shows the inhibitory effect of the *Pediococcus acidilactici* CFS. CFS-50%-W-D-*P. acidilactici* (O.D. 0.078 ± 0.008) inhibited *C. albicans* ATTC 10231 (*p* < 0.05) versus the controls. The presence of 12.5, 25, and 50% concentrations of dextrose CFS supernatants inhibited the growth of clinical *C. albicans*: CFS-12.5%-W-D-*P. acidilactici*, CFS-25%-W-D-*P. acidilactici*, and CFS-12.5%-W-D-*P. acidilactici* had an O.D. of 1.008 ± 0.002, 0.812 ± 0.015, and 0.141 ± 0.064, respectively.

The presence of CFS-25%-W-I-*P. acidilactici* and CFS-50%-W-I-*P. acidilactici* also inhibited the growth of clinical *C. albicans*. The CFS-25%-W-I-*P. acidilactici* and CFS-50%-W-I-*P. acidilactici* changed the O.D. (1.035 ± 0.022 and 0.784 ± 0.035, respectively) versus (O.D. 1.217 ± 0.064) (*p <* 0.05). The inhibitory effect of CFS-50%-N-I-*P. acidilactici* against *C. albicans* (O.D. 1.045 ± 0.008) remained even when subjected to neutralization (Figure 2d).

The thermally-treated CFSs inhibited the growth against *C. albicans* ATCC 10231: 0.063 ± 0.006 and 0.836 ± 0.056 optical densities for CFS-50%-T-I-*P. acidilactici* and CFS-50%-T-D-*P. acidilactici*, respectively (Figure 2e). All CFSs treated thermally inhibited clinical *C. albicans* (Figure 2f). The dextrose CFSs 12.5%-T-D-*P. acidilactici*, CFS-25%-T-D-*P. acidilactici*, and CFS-50%-T-D-*P. acidilactici* inhibited clinical *C. albicans*: O.D. 0.941 ± 0.026, 0.753 ± 0.003, and 0.070 ± 0.003. The inulin supernatants CFS-12.5%-T-I-*P. acidilactici*, CFS-25%-T-I-*P. acidilactici*, and CFS-50%-T-I-*P. acidilactici* inhibited clinical *C. albicans*: O.D. 0.073 ± 0.006, 0.916 ± 0.62, and 0.731 ± 0.003.

Concentrations of 50%, 25%, and 12.5% of CFS were exposed to fresh YPD medium with *Candida* spp. Treatment: without (CFS did not receive any treatment), neutralized (CFS suspensions were adjusted to pH 6.5). Verification code: first, cell-free supernatants. Second, the concentrations of supernatants. Third, the treatment. Fourth, the carbon source, and fifth, probiotic.

Figure 3 shows the antibiofilm activity of the supernatants obtained from *L. rhamnosus* and *P. acidilactici*. CFS-25%-D-*rhamnosus* (77.10%) and CFS-50%-D-*rhamnosus* (88.25%) showed the highest percentages of antibiofilm inhibition against *C. albicans* ATCC 10231 (Figure 3a), while CFS-50%-D-*rhamnosus* (90.83%), CFS-25%-I-*rhamnosus* (89.13%), and CFS-50%-I-*rhamnosus* (91.60%) had the greatest antibiofilm effect against clinical *C. albicans*; no statistical difference was found between them (*p <* 0.05) (Figure 3b). Dextrose and inulin CFSs obtained from *P. acidilactici* showed antibiofilm effects against *C. albicans* ATCC 10231. The percent inhibition ranged from 80.17 to 91.25% for dextrose CFSs and from 75.25 to 85.26% for inulin CFSs (Figure 3c). CFS-25%-D-*P. acidilactici*, CFS-50%-D-*P. acidilactici*, and CFS-50%-I-*P. acidilactici* inhibited biofilm formation against clinical *C. albicans* as follows: 87.99%, 91.17%, and 92.59%, respectively; there was no statistical difference between them (*p <* 0.05) (Figure 3d).

The short-chain fatty acids concentrations are shown in Table 2. Lactate was the dominating metabolite produced by *L. rhamnosus* when supplemented with either dextrose or inulin as a carbon source (8.24 ± 1.36 and 5.87 ± 0.70 mM, respectively) (*p <* 0.05). However, *L. rhamnosus* secreted a greater amount of acetic acid (*p <* 0.05) when supplemented with inulin (7.00 ± 1.26 mM) as a carbon source than dextrose (1.66 ± 0.37 mM). Acetic acid was the dominant SCFA in the *P. acidilactici* CFSs, but dextrose CFS had the highest concentration (17.6 ± 2.54 mM) versus inulin CFS (6.92 ± 0.76 mM) with a statistically significant difference (*p <* 0.05). The lactate concentrations were 9.02 ± 0.40 mM and 5.48 ± 0.17 mM for dextrose and inulin CFSs, respectively; these were statistically different (*p <* 0.05). No statistical difference was found between propionic and butyric acids when comparing the CFSs of *L. rhamnosus* and *P. acidilactici* produced upon supplementation of dextrose or inulin. Propionic and butyric acids were the SCFAs with the lowest concentration in the CFSs.

## 3. Discussion

There are about 200 species in the *Candida* genus, of which 20 species are related to candidiasis in humans; therefore, *Candida* has an important role in public health. *Candida albicans* is the main species related to infections, and this opportunistic pathogen is highly adaptable to human hosts [21]. Studies have demonstrated that *C. albicans* can alter the gut microbiota composition in intestinal bowel disease patients. For which the symbiotic combinations herein presented would be a potential treatment. [25]. The beneficial properties of probiotics are usually evaluated by their effect on human pathogens such as *Candida albicans*. Probiotics are often administrated as living microorganisms due to their benefits, and postbiotics include any substance or product released by or produced through the metabolic activity of probiotics that may exert multiple beneficial effects on the health host directly or indirectly [26]. Antimicrobial activity is a key feature of probiotics, but most research has focused on antimicrobial activity against bacteria. Here, we demonstrated the inhibitory effect against *Candida albicans* of postbiotics produced by *Lacticaseibacillus rhamnosus* and *Pediococcus acidilactici* in combination with inulin-type fructans. The supernatants obtained from the synbiotic combination of *L. rhamnosus* with inulin-type fructans showed growth and biofilm inhibition against *Candida albicans*.

Similarly, Dausset et al., 2020 [27] demonstrated that the supernatant of *Lacticaseibacillus rhamnosus* Lcr35 inhibited the growth of *Candida albicans* ATCC 10231, thus reducing the viability by 3 log10 (CFU/mL). This suggests that the presence of metabolites produced by this probiotic exerts an antimicrobial effect. Some of this inhibition is due to the production of secondary metabolites by *L. rhamnosus* such, as hydrogen peroxide or lactate, that can lyse the fungal cell. However, other mechanisms may be involved in this antimicrobial effect. *L. rhamnosus* GG may produce exopolysaccharides or chitinase to interfere with the hyphal formation of *Candida* spp. [28,29].

The antimicrobial activity of *P. acidilactici* has been demonstrated against bacterial pathogens such as *Listeria monocytogenes*, *Staphylococcus aureus*, and *Salmonella typhimurium* [30]. There is limited literature on the antifungal effect of *Pediococcus acidilactici* and the mechanisms involved. The anticandidal effect of the *P. acidilactici* supernatants demonstrated here is similar to that shown by Crowley et al., 2013: These authors showed that the supernatants from *P. acidilactici* HW01 hindered the growth and biofilm formation of *Candida albicans* [31]. Our results are similar to other studies reporting the antimicrobial effect of different *Lactobacillus* strains that secreted antimicrobial metabolites such as organic acids, hydrogen peroxide, fat metabolites, and bacteriocins. Similarly, the antimicrobial effect of *Pediococcus* spp. against pathogens is due to the secretion of protein-like compounds [32].

Lactic acid bacteria can produce bacteriocins that can modulate pathogens’ growth. In this study, lactate and acetic acid were the dominant metabolites produced by *L. rhamnosus* and *P. acidilactici*. These organic acids can indirectly inhibit pathogens’ growth by decreasing the media’s pH [33]. Furthermore, *L. rhamnosus* produced a higher concentration of acetic acid in the presence of inulin as a carbon source than in the presence of dextrose. Acetic acid-producing bacteria in the intestinal environment can improve colonic health [19]. Therefore, probiotics can modulate gut health in the presence of prebiotics.

The lower antimicrobial effect of the neutralized supernatants of *L. rhamnosus* and *P. acidilactici* in this study may be due to the presence of non-bacteriocin compounds secreted by these probiotics [21]. Only the CFS-50%-N-I-*P. acidilactici* supernatant maintained its growth inhibition effect after being neutralized: This was significantly higher than supernatants without neutralization and suggested that inulin-type fructans may stimulate *P. acidilactici* to secrete a greater quantity of bacteriocin-type compounds. This is consistent with a previous study showing that the combination of prebiotics with probiotic bacteria improves the secretion of bacteriocins [34]. One limitation of the present study was the use of one clinical isolate and one ATCC strain of *C. albicans* only. However, *C. albicans* has displayed a wide range of intraspecies variations and phenotypic properties [35], which may impact the susceptibility to cell-free supernatants obtained from the culture of probiotics.

## 4. Materials and Methods

### 4.1. Candida albicans Culture Conditions

*Candida albicans* was obtained from a patient stool sample and identified by matrix-assisted laser desorption/ionization time-of-flight (MALDI-TOF) mass spectrometry. *Candida albicans* ATCC 10231 was acquired from the American Type Culture Collections and was used as a control. Yeast-peptone-dextrose (YPD) (SIGMA-Aldrich^®^, St. Louis, MO, USA) was the culture medium for *Candida* at 37 °C.

### 4.2. Probiotics Cultivations and Cell-Free Supernatants Obtention

*Lacticaseibacillus rhamnosus* NH001 and *Pediococcus acidilactici* MA18/5M, previously identified by MALDI-TOF, were used as probiotics and were cultured for cell-free supernatants. These probiotics were cultured in Man–Rogosa–Sharpe (MRS) culture medium (DIFCOTM, Le Pont de Claix, France) at 37 °C. Cell-free supernatants were obtained following the methodology reported by García-Gamboa et al., 2022 [36]. *L. rhamnosus* and *P. acidilactici* were incubated in MRS broth at 37 °C for 24 h. In addition, a modified MRS culture medium was prepared by replacing dextrose with inulin-type fructans as a carbon source. After incubation, the culture medium was centrifugated at 2800× *g* for 10 min and 4 °C to obtain cell-free supernatants. The cell-free supernatants were filtered using a 0.45 μm pore size sterile filter (Corning^®^, Corning, NY, USA). The additional supernatant samples were neutralized to pH 6.5 with a sterile solution of 1 N NaOH (pH 6.5) to determine if the inhibitory effect was due to the acidity of the cultured medium. Other supernatant samples were heated at 121 °C for 15 min to assess the resistance of the metabolites to heat [21].

### 4.3. Growth Inhibition of Candida albicans

Cell-free supernatants from *L. rhamnosus* and *P. acidilactici* were exposed to *Candida albicans* to evaluate their inhibitory effect. The broth microdilution analysis in 96-well microtiter plates was implemented as reported by the Clinical and Laboratory Standards Institute (CLSI, 2012). *Candida* spp. was grown in YPD broth at 37 °C for 24 h. Next, 100 µL of this sample with 1 × 10^5^ UFC/mL (0.2 at a 600 nm optical density) of *Candida* spp. was then placed in 5 mL of fresh YPD broth. The *L. rhamnosus* and *P. acidilactici* cell-free supernatants were serially diluted with two-fold dilutions in YPD broth in a 96-well microtiter plate. Additionally, 100 µL of *C. albicans* ATCC 10231 and clinical *C. albicans* were placed in the presence of different concentrations of cell-free supernatants. The growth inhibition of *Candida albicans* was analyzed through spectrophotometry (600 nm optical density), and measurements were taken every hour for 18 h.

### 4.4. Antibiofilm Activity against Candida albicans

Samples of a 16-h culture medium of clinical *C. albicans* and *C. albicans* ATCC (100 µL (1 × 10^5^ CFU/mL)) were added to a 5 mL of fresh YPD medium. After 18 h of incubation, 150 µL was added in triplicate to a 96-well microplate in the presence of 25 µL, 50 µL, and 100 µL of the *L. rhamnosus* and *P. acidilactici* cell-free supernatants. The 96-well microplate was incubated at 37 °C for 24 h. Wells containing only *Candida* spp. with YPD medium were used as a control, and chloramphenicol was used as a negative control. After incubation, the well contents were removed, and the wells were washed with a phosphate-buffered saline solution (pH 7.0). In addition, 100 µL of methanol (99% *v*/*v*) was placed in the wells for 15 min, and the methanol was removed. The wells were then dried under sterile air for 5 min, and 100 µL of crystal violet was added to the wells for 15 min. The stain was then removed, and the wells were gently washed with water. The wells were allowed to dry for 5 min, and the absorbance at 620 nm was measured with a microplate spectrophotometer [36]. The biofilm inhibition of *Candida* spp. was calculated with the following equation:(1)% Biofilm reduction=1−(O.D.1O.D.2)×100%
where:*O*.*D*._1_ = absorbance of wells containing cell-free supernatant and *Candida* spp.*O*.*D*._2_ = absorbance of wells containing only *Candida* spp. (control).

### 4.5. Determination of SCFAs in Cell-Free Supernatants

The short-chain fatty acids (SCFAs) concentrations were measured by gas chromatography and flame ionization (GC 2010, Shimatzu^®^, Kyoto, Japan) using the method reported by Femia et al., 2002 with some modifications [37]. Here, 250 µL of each supernatant was acidified with 25 µL phosphoric acid solution (0.5 M). The short-chain fatty acids were extracted by shaking with 250 µL of diethyl ether and then centrifugated (9400× *g* 3 min). Next, 5 µL of the ether phase (5 µL) was directly injected onto a column (HP-FFAPP 30 m × 0.250 mm × 0.25 mm, Agilent JW GC columns, Santa Clara, CA, USA) at 180 °C using N_2_ as the carrier gas for 10 min; the temperature for detection and injection was 230 °C.

### 4.6. Statistical Analysis

All of the tests were performed in triplicate, and the data were represented as mean ± SD (mean ± standard deviation). The mean values of the optical density measurements and SCFA concentrations were compared between experimental treatments using one-way ANOVA, and *p <* 0.05 was considered significant. GraphPad prism 8.4.3 (GraphPad Software Inc., La Jolla, CA, USA) was used for analysis.

## 5. Conclusions

In summary, this study showed that a synbiotic combination of *Lacticaseibacillus rhamnosus* and *Pediococcus acidilactici* with inulin-type fructans displayed growth and biofilm inhibition against *Candida albicans*. These synbiotic formulations could be a promising alternative to the use of antifungal drugs. However, further in vivo studies are needed to confirm these findings.

## Figures and Tables

**Figure 1 antibiotics-11-01135-f001:**
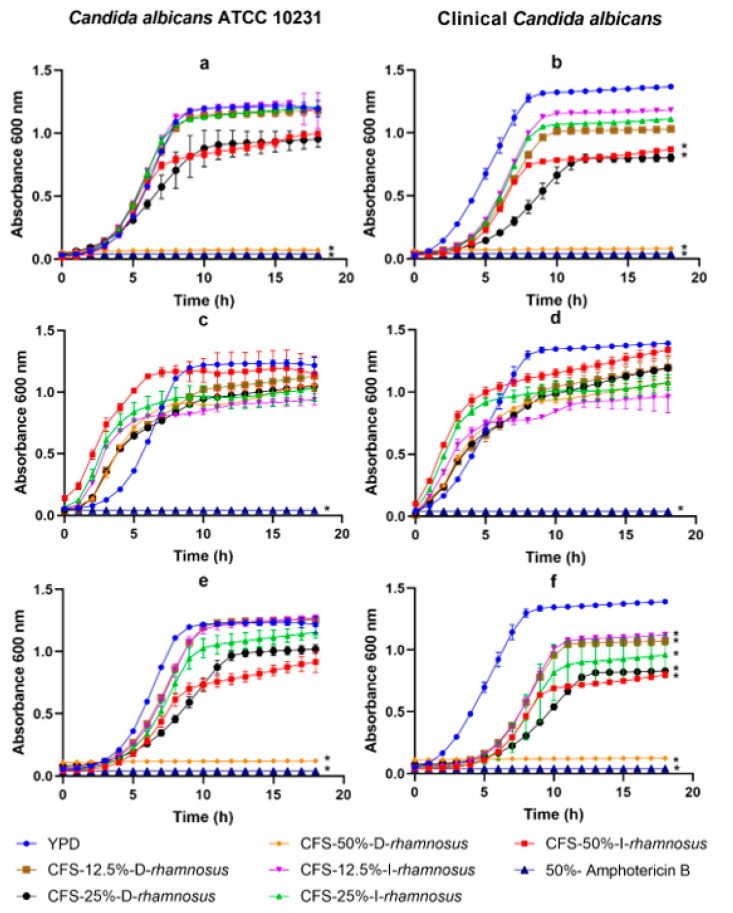
*Candida* spp. growth kinetics in the presence of different cell-free supernatant concentrations obtained from *Lacticaseibacillus rhamnosus*: (**a**,**b**) supernatants without treatment, (**c**,**d**) neutralized supernatants, and (**e**,**f**) supernatants at 121 °C. The values are reported in optical density (O.D.) as mean ± SD of three replicates. (*) means statistical difference compared to control (*p <* 0.05).

**Figure 2 antibiotics-11-01135-f002:**
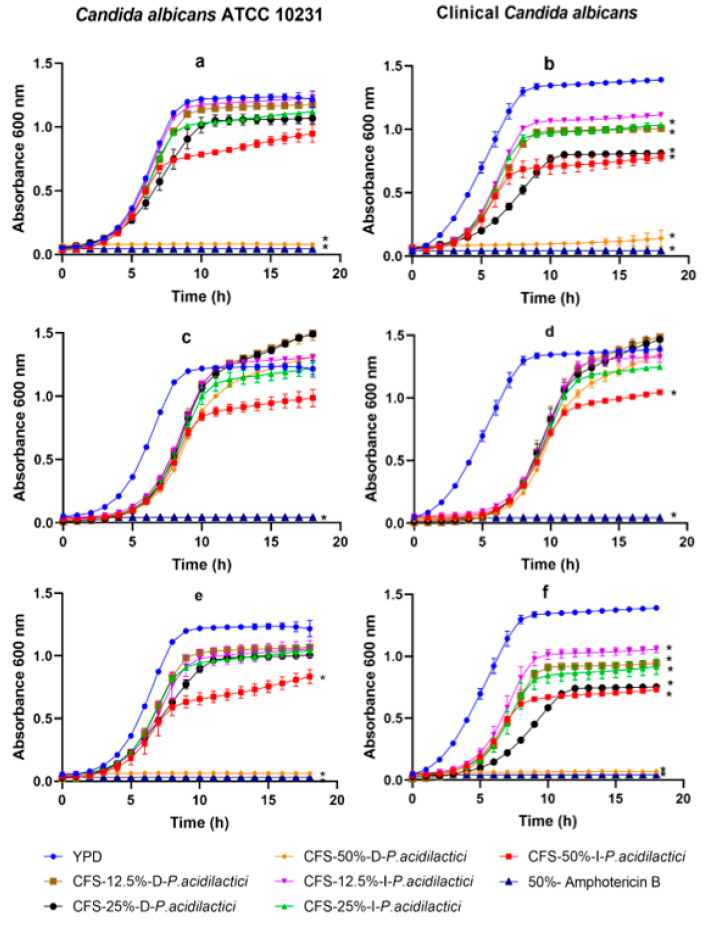
*Candida* spp. growth kinetics in the presence of different cell-free supernatant concentrations obtained from *Pediococcus acidilactici*: (**a**,**b**) Supernatants without treatment, (**c**,**d**) neutralized supernatants, and (**e**,**f**) supernatants at 121 °C. The values are reported in optical density (O.D.) as mean ± SD of three replicates. (*) means statistical difference versus control (*p <* 0.05).

**Figure 3 antibiotics-11-01135-f003:**
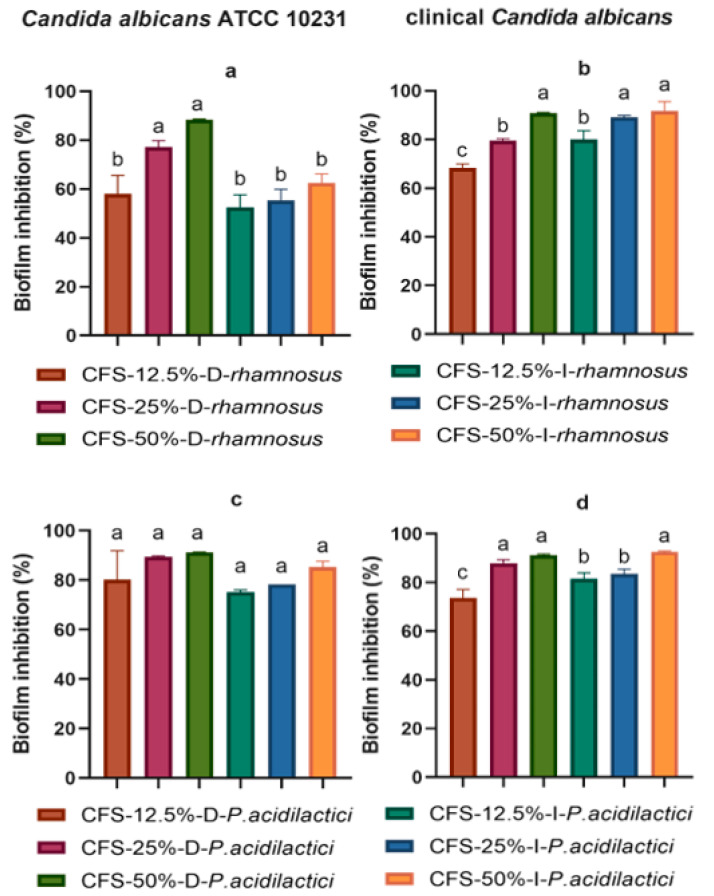
Biofilm inhibition of *Candida* spp. in the presence of different concentrations of cell-free supernatants: (**a**,**b**) are the inhibition of cell-free supernatants obtained from *Lacticaseibacillus rhamnosus*. (**c**,**d**) are the inhibition of cell-free supernatants obtained from *Pediococcus acidilactici*. The values are reported in optical density (O.D.) as mean ± SD of three replicates. Different lowercase letters indicate significant differences between treatments (*p* < 0.05).

**Table 1 antibiotics-11-01135-t001:** Description of the supernatants obtained from probiotic bacteria.

Probiotic:				*L. rhamnosus*	*P. acidilactici*
Cell Free Supernatant	Concentration Percentage (%)	Postbiotics Treatment	Carbon Source	Verification Code	
CFS	50	Without	Inulin	CFS-50%-W-I-*rhamnosus*	CFS-50%-W-I-*P. acidilactici*
	25			CFS-25%-W-I-*rhamnosus*	CFS-25%-W-I-*P. acidilactici*
	12.5			CFS-12.5%-W-I-*rhamnosus*	CFS-12.5%-W-I-*P. acidilactici*
	50		Dextrose	CFS-50%-W-D-*rhamnosus*	CFS-50%-W-D-*P. acidilactici*
	25			CFS-25%-W-D-*rhamnosus*	CFS-25%-W-D-*P. acidilactici*
	12.5			CFS-12.5%-W-D-*rhamnosus*	CFS-12.5%-W-D-*P. acidilactici*
	50	Neutralized	Inulin	CFS-50%-N-I-*rhamnosus*	CFS-50%-N-I-*P. acidilactici*
	25			CFS-25%-N-I-*rhamnosus*	CFS-25%-N-I-*P. acidilactici*
	12.5			CFS-12.5%-N-I-*rhamnosus*	CFS-12.5%-N-I-*P. acidilactici*
	50		Dextrose	CFS-50%-N-D-*rhamnosus*	CFS-50%-N-D-*P. acidilactici*
	25			CFS-25%-N-D-*rhamnosus*	CFS-25%-N-D-*P. acidilactici*
	12.5			CFS-12.5%-N-D-*rhamnosus*	CFS-12.5%-N-D-*P. acidilactici*
	50	Thermal	Inulin	CFS-50%-T-I-*rhamnosus*	CFS-50%-T-I-*P. acidilactici*
	25			CFS-25%-T-I-*rhamnosus*	CFS-25%-T-I-*P. acidilactici*
	12.5			CFS-12.5%-T-I-*rhamnosus*	CFS-12.5%-T-I-*P. acidilactici*
	50		Dextrose	CFS-50%-T-D-*rhamnosus*	CFS-50%-T-D-*P. acidilactici*
	25			CFS-25%-T-D-*rhamnosus*	CFS-25%-T-D-*P. acidilactici*
	12.5			CFS-12.5%-T-D-*rhamnosus*	CFS-12.5%-T-D-*P. acidilactici*

**Table 2 antibiotics-11-01135-t002:** Short-chain fatty acid produced by probiotics on dextrose and inulin fermentation.

	*L. rhamnosus*		*P. acidilactici*	
Lactate and SCFAs	Dextrose	Inulin	Dextrose	Inulin
Lactate	8.24 ± 1.36 a	5.87 ± 0.70 b	9.02 ± 0.40 a	5.48 ± 0.17 b
Acetic acid	1.66 ± 0.37 b	7.00 ± 1.26 a	17.6 ± 2.54 a	6.92 ± 0.76 b
Propionic acid	0.21 ± 0.04 a	0.23 ± 0.06 a	0.66 ± 0.067 a	0.20 ± 0.03 a
Butyric acid	1.83 ± 0.23 a	1.89 ± 0.20 a	1.80 ± 0.04 a	1.67 ±0.02 a

SCFAs: Short-chain fatty acids. Different lowercase letters indicate significant differences between the carbon sources used by each microorganism for metabolite content. The values are reported in mM as mean ± SD of three replicates (*p <* 0.05).

## Data Availability

All data are available upon request.

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
