# Peer review of "Anticandidal and Antibiofilm Effect of Synbiotics including Probiotics and Inulin-Type Fructans"

_antibiotics, 2022, doi:10.3390/antibiotics11081135_

Round 1

Reviewer 1 Report

Overall the manuscript is more suited for short and interesting reports. It is my evaluation that the findings presented are incremental at best and more suited for elsewhere

Author Response

Thank you for the valuable comments and observations made. Regarding your suggestion that this work would stand out more elsewhere, we believe that this article can fit into the special issue to which we were cordially invited, and we are grateful for that.

Reviewer 2 Report

Not all the focuses of the work are well organized in the introduction

LINE 46:

 Introduce this sentence: “Furthermore, one of the great challenges to the use of conventional antimicrobials is the multifactorial nature of biofilm development, as it indicates the need for multi-targeted or combinatorial treatment. Azoles, polyenes and echinocandins are common antifungal drugs, but they have  some adverse effects such as mutation and drug resistance with Candida albicans [16]. Several studies have focused on the development of antimicrobials against bacteria ,with better results, especially when the antimicrobial compounds have been functionalized into nanoparticles that mimic extracellular structures, as release vectors for antibacterial agents. (Citation: Montone, A.M.I.; Papaianni, M.; Malvano, F.; Capuano, F.; Capparelli, R.; Albanese, D. Lactoferrin, Quercetin, and Hydroxyapatite Act Synergistically against Pseudomonas fluorescens. Int. J. Mol. Sci. 2021, 22, 9247. https:// doi.org/10.3390/ijms22179247). Also Hong et al, 2019 [19] reported that pullulan nanoparticles enhanced the antibacterial  properties of Lactobacillus plantarum because this probiotic produced more plantaricin a  natural antibacterial peptideIn the literature, substances with growth inhibition activity have also been reported for fungal microorganisms. A previous study suggested that Lacticaseibacillus rhamnosus can alter the susceptibility of Candida albicans to antifungal drugs [11]. Another study demonstrated that cell-free supernatants (CFS) obtained from Pediococcus acidilactici inhibited Candida albicans  [18].  It is now ascertained that fatty acids possess anti-infective activities, exerted through various physical, chemical, or biological mechanisms. Recently, many researchers have directed their interests towards the study of antimicrobial lipids and numerous studies demonstrated that fatty acids inhibit or kill a wide spectrum of pathogens, suppress the expression of Quorum Sensing-regulated genes, reduce swarming motility, adhesion, and virulence, and directly induce biofilm dispersion”. (Citation: Papaianni, M.; Ricciardelli, A.; Casillo, A.; Corsaro, M.M.; Borbone, F.; Della Ventura, B.; Velotta, R.; Fulgione, A.; Woo, S.L.; Tutino, M.L.; et al. The Union Is Strength: The Synergic Action of Long Fatty Acids and a Bacteriophage against Xanthomonas campestris Biofilm. Microorganisms 2021, 9, 60. https:// doi.org/10.3390/microorganisms90 10060)

Author Response

Done in the main document. Lines: 48-67

Reviewer 3 Report

Nice piece of research. Need more clinical approval. 

Author Response

It is not the focus of this work, but in a future, it is considered to investigate the clinical approval

Reviewer 4 Report

This study aimed to evaluate the inhibition of Candida growth and the antibiofilm effect of cell-free supernatants obtained from the culture of Lacticaseibacillus rhamnosus and Pediococcus acidilactici in MRS broth with inulin. The authors observed that supernatants inhibited the growth and formation of biofilms against Candida albicans and suggested that the synbiotic combination would be a promising alternative to the use of antifungals. Below are some suggestions and questions:

- This is an in vitro study where only one clinical isolate and one ATCC strain of C. albicans were used. There are intraspecies variations that can interfere with the susceptibility of an isolate. Authors should mention the limitations of the study in the discussion section;

- Symbiotics are used by ingestion. Considering the authors suggested that the symbiotic combination could be an alternative to the use of antifungals, in which clinical cases of candidiasis would there be a potential application for it?

- The symbols used in figures 1 and 2 could be better visualized if they were in different colors;

- What is the origin of Lacticaseibacillus rhamnosus NH001 and Pediococcus acidilactici MA18/5M strains?

Author Response

- This is an in vitro study where only one clinical isolate and one ATCC strain of C. albicans were used. There are intraspecies variations that can interfere with the susceptibility of an isolate. Authors should mention the limitations of the study in the discussion section;

Answer [this part was placed in the discussion section. Lines: 215-219]:

One limitation of the present study was the use of one clinical isolate and one ATCC strain of C. albicans only. However, C. albicans has displayed a wide range of intraspecies variations and phenotypic properties [35], which may impact the susceptibility to cell-free supernatants obtained from the culture of probiotics.

- Symbiotics are used by ingestion. Considering the authors suggested that the symbiotic combination could be an alternative to the use of antifungals, in which clinical cases of candidiasis would there be a potential application for it?

Answer [this part was placed in the discussion section. Lines: 168-170]:

Studies have demonstrated that C. albicans can alter the gut microbiota composition in intestinal bowel disease patients. for which the symbiotic combinations herein presented would be a potential treatment. 

- The symbols used in figures 1 and 2 could be better visualized if they were in different colors;

 Answer [Done in the main document. Pages:4-6]

- What is the origin of Lacticaseibacillus rhamnosus NH001 and Pediococcus acidilactici MA18/5M strains?

[Answer] Both strains were acquired from DuPontTM .

Round 2

Reviewer 1 Report

Thanks a lot for your reply to my queries. I wish you best of luck.

Reviewer 4 Report

The authors accepted all suggestions. The revised version of the manuscript is suitable for publication.